

# Characterization of antimicrobial resistance genes in *Haemophilus parasuis* isolated from pigs in China

Yongda Zhao[1], Lili Guo[2], Jie Li[1], Xianhui Huang[1] and Binghu Fang[1]

[1] College of Veterinary Medicine, South China Agricultural University, Guangzhou, Guangdong, China
[2] Qingdao Yebio Biological Engineering Co., Ltd, Qingdao, Shandong, China

## ABSTRACT

**Background**. *Haemophilus parasuis* is a common porcine respiratory pathogen that causes high rates of morbidity and mortality in farmed swine. We performed a molecular characterization of antimicrobial resistance genes harbored by *H. parasuis* from pig farms in China.

**Methods**. We screened 143 *H. parasuis* isolates for antimicrobial susceptibility against six fluoroquinolone antibiotics testing by the broth microdilution method, and the presence of 64 antimicrobial resistance genes by PCR amplification and DNA sequence analysis. We determined quinolone resistance determining region mutations of DNA gyrase (*gyrA* and *gyrB*) and topoisomerase IV (*parC* and *parE*). The genetic relatedness among the strains was analyzed by pulsed-field gel electrophoresis.

**Results**. Susceptibility test showed that all isolates were low resistance to lomefloxacin (28.67%), levofloxacin (20.28%), norfloxacin (22.38%), ciprofloxacin (23.78%), however, high resistance levels were found to nalidixic acid (82.52%) and enrofloxacin (55.94%). In addition, we found 14 antimicrobial resistance genes were present in these isolates, including *bla*$_{TEM-1}$, *bla*$_{ROB-1}$, *ermB*, *ermA*, *flor*, *catl*, *tetB*, *tetC*, *rmtB*, *rmtD*, *aadA1*, *aac(3′)-llc*, *sul1*, and *sul2* genes. Interestingly, one isolate carried five antibiotic resistance genes (*tetB, tetC, flor, rmtB, sul1*). The genes *tetB*, *rmtB*, and *flor* were the most prevalent resistance genes in *H. parasuis* in China. Alterations in the *gyrA* gene (S83F/Y, D87Y/N/H/G) were detected in 81% of the strains and *parC* mutations were often accompanied by a *gyrA* mutation. Pulsed-field gel electrophoresis typing revealed 51 unique patterns in the isolates carrying high-level antibiotic resistance genes, indicating considerable genetic diversity and suggesting that the genes were spread horizontally.

**Discussion**. The current study demonstrated that the high antibiotic resistance of *H. parasuis* in piglets is a combination of transferable antibiotic resistance genes and multiple target gene mutations. These data provide novel insights for the better understanding of the prevalence and epidemiology of antimicrobial resistance in *H. parasuis*.

Corresponding author
Binghu Fang, fangbh@scau.edu.cn

## INTRODUCTION

*Haemophilus parasuis* is the etiological agent of Glässer's disease that causes significant morbidity and mortality as well as economic losses in the global pig industry (*Oliveira & Pijoan, 2004*). Antimicrobial therapy is used to prevent and control this infection even though antimicrobial agents are also used for growth promotion in pigs (*Lancashire et al., 2005*). However, extended agricultural use of antibiotics poses a risk for selecting antibiotic resistant pathogens, and antibiotic resistance in *H. parasuis* is increasing (*Aarestrup, Seyfarth & Angen, 2004*; *De la Fuente et al., 2007*; *Markowska-Daniel et al., 2010*; *Walsh & Fanning, 2008*; *Wissing, Nicolet & Boerlin, 2001*; *Xu et al., 2018*). In China, the resistance rate of *H. parasuis* to antimicrobials is also increasing, resulting in limited therapeutic choices (*Zhou et al., 2010*).

Increases in antibiotic resistance among bacteria is most often the result of antibiotic resistance gene (ARG) transfer mediated by mobile DNA elements such as plasmids, transposons and integrons in Gram-negative bacteria (*Lancashire et al., 2005*; *San Millan et al., 2007*). A long history of antibiotic use in the swine industry has generated a strong selective pressure for resistance transfer mediated by plasmids and transposons within and between bacterial species. Plasmids play a key role in this process by acting as vehicles for horizontal gene transfer (*San Millan et al., 2016*). The most prominent ARG types associated with resistance in *H. parasuis* include $bla_{ROB-1}$, *tetB, tetL, qnrA1, qnrB6, aac (6')-Ib-cr, lnu(C)* and *flor* (*Dayao et al., 2016*; *Guo et al., 2011*; *Kehrenberg et al., 2005*; *Lancashire et al., 2005*; *Li et al., 2015*; *San Millan et al., 2007*). In China, $bla_{ROB-1}$, *qnr*A1, *qnr*B6, *aac* (6')-*Ib-cr, lnu*(C) and *flor* have been identified in *H. parasui* s (*Guo et al., 2012*; *Guo et al., 2011*; *Li et al., 2015*). Horizontal gene transfer of ARG-carrying mobile elements and vertical gene transfer by the proliferation of ARG hosts facilitate resistance spread (*Xu et al., 2018*). Moreover, quinolone resistance determining region mutations (QRDR) of *gyrA* and *parC* were related to resistance. Therefore, studying ARG fates and their horizontal and vertical transfer-related elements and QRDRs can provide a comprehensive insight into resistance mechanisms.

*H. parasuis* is one of the most important respiratory pathogens in pigs (*Guo et al., 2012*; *Zhang et al., 2014*; *Zhou et al., 2010*), so more information is needed on the characterization of resistance genes associated with the increase in antibiotic resistance for this bacterium. In the present study, we examined resistance determinants, QRDRs and genetic relatedness in *H. parasuis* strains from pig farms in China.

## MATERIALS AND METHODS

### Bacterial strains

We isolated 143 *H. parasuis* strains from different diseased swine suffering polyserositis, pneumonia or meningitis between February 2014 and March 2017 in China. All strains were isolated from lung, brain, heart blood, pericardial effusion, pleural effusion, peritoneal effusion and joint fluid by aseptic inoculation ring, and cultured on tryptic soy agar (TSA) or tryptic soy broth (TSB) (Becton Dickinson, Owings Mills, MD, USA) containing 10 µg/ml nicotinamide adenine dinucleotide (NAD; Sigma, St. Louis, MO, USA) and

**Table 1  Antibiotic resistance gene testing of the *H. parasuis* isolates in this study.**

| Antibiotic | Resistance genes | Primers |
|---|---|---|
| quinolones | *qepA, qnrA, qnrB, qnrC, qnrD, qnrS, oqxAB, aac(6′)-Ib-cr* | *Cavaco et al. (2009), Yang et al. (2017), Zhao et al. (2010)* |
| β-lactams | *bla*$_{TEM}$ − 1, *bla*$_{ROB−1}$, *SHV, CTX-M-1G, CTX-M-9G, CTX-M-2G, CTX-M-64, CTX-M-25 DHA, VIM-1, VIM-2, SPM-1, CMY-2, npmA, OXA, NDM, KPC, IMP, SPM, FOX* | *Grobner et al. (2009), Liu et al. (2007), San Millan et al. (2007), Weill et al. (2004)* |
| tetracyclines | *tetA, tetB, tetC, tetD, tetE, tetG, tetH, tetL-1, tetL-2* | *De Gheldre et al. (2003), Matter et al. (2007), Miranda, Rodriguez & Galan-Vidal (2009)* |
| aminoglycosides | *rmtB, rmtC, armA, rmtA, rmtD, aadB[ant(2′)-la], aacC2 [aac(3)-Iic], aacC4 [aac(3)-Iva], aadA1,aac(6)-31* | *Doi & Arakawa (2007), Matter et al. (2007)* |
| macrolides | *ermA, ermB, ermC, mefA/E* | *Hou et al. (2013), Matter et al. (2007), Sutcliffe et al. (1996)* |
| chloramphenicol | *catl, cmlA, flor, cfr* | *Maka & Popowska (2016), Wang et al. (2015)* |
| sulfonamides | *sul1, sul2, sul3, dfrA1, dfrB* | *Matter et al. (2007)* |
| integrase gene | *intl1, intl2, intl3* | *Shibata et al. (2003)* |

5% bovine serum (Gibco, Auckland, New Zealand). Plates were incubated at 37 °C for 24–48 h. All isolates were identified by PCR (*Angen et al., 2007*). The study was approved (No.2014-025).

## Fluoroquinolone antimicrobial susceptibility testing

Nalidixic acid, ciprofloxacin, levofloxacin, enrofloxacin, norfloxacin and lomefloxacin were obtained from the National Institute for the Control of Pharmaceutical and Biological Products, Beijing, China. Minimal inhibitory concentrations (MIC) were determined in fastidious medium consisting of TSB with 5% bovine serum and 10 μg/mL NAD in 96-well microtiter plates. All plates were inoculated following the guidelines of the Clinical and Laboratory Standards Institute (CLSI) using *Haemophilus influenzae* and *Haemophilus parainfluenzae* M02 and M07(*CLSI, 2015*). The plates were incubated in an atmosphere containing 5% $CO_2$ at 37 °C for 24 h. The MIC value was defined as the lowest concentration resulting in no visible bacterial growth. The reference strains *H. influenzae* ATCC 49247 and *Escherichia coli* ATCC 25922 served as quality controls for MIC determinations.

## ARGs and integrons detection

DNA was extracted from whole organisms using the quick boiling method (*Sambrook & Russell, 2001*). PCR assays were used to screen for the presence of 64 ARG types including resistance to quinolones, β-lactams, macrolides, tetracycline, aminoglycosides, chloramphenicol, sulfonamides as well as for the integrase gene (Table 1). Purified PCR products were directly sequenced from both ends or cloned into plasmid vector pMD18-T, and then sequenced. DNA sequence similarity searches were performed against the GenBank database using BLAST software to confirm gene identity.

## Detection of mutations in QRDRs of *gyr*A, *gyr*B, *par*C, and *par*E

Mutations in the quinolone resistance determining regions (QRDR) mutations in the *gyrA, gyrB, parC* and *parE* genes were identified after DNA sequencing of PCR products generated with the primers listed in Table 2.

**Table 2  PCR primer sequences used to amplify QRDR genes.**

| Gene | Primers | Sequence (5′-3′) | Size (bp) | Reference |
|------|---------|-------------------|-----------|-----------|
| *gyrA* | GyrA-F | AGCGTTACCAGATGTGCGAGATG | 620 | This study |
|      | GyrA-R | TTGCCACGACCTGTACGATAAGC |  |  |
| *gyrB* | GyrB-F | TACATACGCTGTAGGTTCAAGGA | 500 | This study |
|      | GyrB-R | CAAGATAATACGGAAATGGAGC |  |  |
| *parC* | ParC-F | AACTTCAACATTACCACTTAGCCCTCG | 1,445 | This study |
|      | ParC-R | TACCTCACCAAGCCTCGCCATCT |  |  |
| *parE* | ParE-F | CGATAATTCCCTTGAAGTCGTTG | 609 | This study |
|      | ParE-R | ATTGATCTGCTCGCCACCCTCTG |  |  |

## Pulsed-field gel electrophoresis

Genetic relatedness of *H. parasuis* strains carrying ARGs was determined by pulsed field electrophoresis (PFGE) of *CpoI* - (TaKaRa, Beijing, China) digested genomic DNA samples (*Zhang et al., 2011*). PFGE typing used a CHEF Mapper electrophoresis system (BioRad, Hercules, CA, USA) with 2.16–63.8 sfor 21 h. *Salmonella enterica* serovar Braenderup H9812 DNA digested with *CpoI* was used for a size standard. Interpretation of the PFGE patterns was accomplished using BioNumerics 6.6 software (Applied Maths, Sint-Martens-Latem, Belgium) (*Tenover et al., 1995*).

# RESULTS

## Bacterial strains analysis

In the current study, 143 *H. parasuis strains* were isolated and 73 carried antibiotic resistance genes. Information on isolation site, isolation time and resistance gene content are listed in Table 3.

## Fluoroquinolone antimicrobial susceptibility testing

The results of the fluoroquinolone antimicrobial susceptibility of 143 *H. parasuis* isolates are listed in Supplemental Information 1. It showed that 82.52% and 55.94% of all isolates were resistant to nalidixic acid and enrofloxacin, respectively. Resistance of lomefloxacin, levofloxacin, norfloxacin, ciprofloxacin were 28.67%, 20.28%, 22.38%, 23.78%, respectively.

## ARG and integron prevalence and detection

We examined 143 *H. parasuis* strains and 16 (11.2%) carried β-lactamases including $bla_{TEM-1}$ and $bla_{ROB-1}$. Tetracycline resistant strains carried *tetB* and *tetC*. There were two isolates (1.40%) also yielded the erythromycin resistance genes: 1 for *ermA*, and 1 for *ermB*. A higher proportion (16.1%) carried chloramphenicol resistance genes including 10 *catl* and 13 *flor*. Aminoglycoside resistance was also high (11.9%) and included the genes *rmtB*, *rmtD*, *aadA1* and *aac(3′)*- ll*c*. The sulfonamide resistance genes were represented by *sul1* and *sul2* and were found in 9 (6.3%) and 2 (1.4%) of the isolates, respectively (Table 4).

The resistance gene patterns were diverse and 39 isolates carried one gene, 24 carried two and nine isolates carried three genes. Interestingly, strain HP142 carried five genes

**Table 3** List *H. parasuis* strains with their separation site, date, organ and resistance gene.

| Isolates | Separation site | Date | Organ | Resistance gene |
|----------|-----------------|------|-------|-----------------|
| HP001 | Fujian | 2016 | lung | *rmtB* |
| HP008 | Fujian | 2015 | nasal cavity | *tetB* |
| HP011 | Meizhou | 2014 | pericardial effusion | *sul2*+ *bla*$_{ROB-1}$ |
| HP012 | Jinan | 2017 | lung | *bla*$_{TEM-1}$ |
| HP013 | Zengcheng | 2016 | nasal cavity | *catl1*+*tetB*+ *bla*$_{ROB-1}$ |
| HP016 | Laiyang | 2015 | brain | *ermA* |
| HP017 | Dongguan | 2015 | joint fluid | *tetB*+*tetC* |
| HP018 | Qingdao | 2016 | lung | *ermB* |
| HP019 | Hebei | 2017 | lung | *sul2*+*tetB* |
| HP020 | Jilin | 2016 | lung | *tetB* |
| HP022 | Huadou | 2015 | lung | *bla*$_{TEM-1}$ |
| HP025 | Zengcheng | 2016 | joint fluid | *catl1*+*tetB*+*aac(3′)-IIc* |
| HP026 | Guangxi | 2014 | lung | *tetB*+*flor* |
| HP029 | Guangxi | 2015 | heart blood | *tetB*+*flor*+ *aac(3′)-IIc* |
| HP032 | Chengde | 2014 | joint fluid | *aadA1* |
| HP035 | Guangxi | 2017 | heart blood | *catl1* |
| HP037 | Fujian | 2015 | nasal cavity | *rmtB* |
| HP039 | Hebei | 2015 | pericardial effusion | *aac(3′)-IIc* |
| HP040 | Jiangmen | 2016 | lung | *sul1*+ *aac(3′)-IIc* |
| HP044 | Jiangsu | 2014 | lung | *tetB* |
| HP050 | Jiangsu | 2016 | pleural effusion | *tetB* |
| HP051 | Jiangsu | 2014 | heart blood | *tetC* |
| HP053 | Yunnan | 2016 | lung | *tetB*+*flor* |
| HP054 | Guangzhou | 2016 | lung | *tetB* |
| HP056 | Zhucheng | 2017 | lung | *rmtB*+*sul1* |
| HP059 | Guangxi | 2016 | lung | *catl1* +*tetB* |
| HP060 | Guangxi | 2015 | heart blood | *tetC*+*flor* |
| HP061 | Qingdao | 2016 | lung | *rmtB* |
| HP063 | Hebei | 2015 | lung | *bla*$_{TEM-1}$ |
| HP065 | Qingyuan | 2015 | lung | *rmtB*+ *bla*$_{TEM-1}$ |
| HP066 | Guangxi | 2016 | lung | *sul1*+ *aac(3′)-IIc* |
| HP067 | Qingdao | 2015 | lung | *tetB*+*aadA1* |
| HP068 | Qingyuan | 2015 | heart blood | *tetB* |
| HP069 | Hunan | 2017 | heart blood | *flor* |
| HP071 | Hebei | 2015 | lung | *tetB* |
| HP072 | Zhucheng | 2017 | nasal cavity | *tetB* |
| HP073 | Zhuhai | 2016 | joint fluid | *bla*$_{ROB-1}$ |
| HP075 | Fujian | 2014 | pericardial effusion | *sul1* |
| HP076 | Henan | 2017 | heart blood | *tetB* |
| HP078 | Henan | 2015 | lung | *rmtB*+ *bla*$_{TEM-1}$ |
| HP079 | Jining | 2016 | lung | *rmtB* |
**Table 3** (*continued*)

| Isolates | Separation site | Date | Organ | Resistance gene |
|---|---|---|---|---|
| HP080 | Jinan | 2014 | joint fluid | $rmtB+sul1$ |
| HP082 | Qingdao | 2016 | lung | $tetB$ |
| HP085 | Liaoning | 2016 | lung | $tetB$ |
| HP091 | Shaoguan | 2017 | pericardial effusion | $catl1+tetB$ |
| HP094 | Hebei | 2015 | lung | $catl1+tetB+ bla_{ROB-1}$ |
| HP095 | Huadou | 2017 | lung | $catl1+tetB$ |
| HP096 | Zhengzhou | 2014 | heart blood | $rmtB$ |
| HP097 | Hunan | 2016 | pericardial effusion | $tetB$ |
| HP098 | Hebei | 2014 | lung | $tetB$ |
| HP102 | Anyang | 2015 | lung | $bla_{ROB-1}+aadA1$ |
| HP103 | Hunan | 2017 | lung | $catl1+tetB+flor$ |
| HP104 | Jiangsu | 2016 | joint fluid | $flor+aadA1$ |
| HP108 | Guangxi | 2016 | lung | $tetB+flor+rmtB$ |
| HP109 | Zhaoqing | 2017 | heart blood | $tetB$ |
| HP111 | Jiangxi | 2015 | lung | $rmtB$ |
| HP112 | Sihui | 2016 | heart blood | $tetB+bla_{ROB-1}$ |
| HP113 | Henan | 2015 | lung | $tetB+tetC+flor$ |
| HP116 | Boluo | 2014 | lung | $bla_{TEM-1}$ |
| HP117 | Hebei | 2017 | heart blood | $rmtD+rmtB+bla_{TEM-1}$ |
| HP118 | Huizhou | 2016 | lung | $rmtB+aac(3')-IIC$ |
| HP120 | Hebei | 2016 | lung | $tetB$ |
| HP121 | Hebei | 2016 | heart blood | $flor$ |
| HP123 | Anhui | 2015 | lung | $flor$ |
| HP127 | Jilin | 2015 | joint fluid | $flor$ |
| HP131 | Yunnan | 2014 | heart blood | $sul1$ |
| HP133 | Huizhou | 2016 | lung | $catl1+rob-1$ |
| HP134 | Zhucheng | 2014 | lung | $rmtB+sul1$ |
| HP135 | Shaoguan | 2016 | pleural effusion | $tetB$ |
| HP137 | Yangzhou | 2015 | lung | $catl1+tetB+ bla_{TEM-1}$ |
| HP140 | Conghua | 2014 | heart blood | $rmtB+ bla_{TEM-1}$ |
| HP141 | Yangzhou | 2017 | lung | $rmtB+sul1$ |
| HP142 | Henan | 2016 | lung | $tetB+tetC+flor+rmtB+sul1$ |

*tetB, tetC, flor, rmtB* and *sul1*. Overall, *tetB*, *rmtB* and *flor* were the most prevalent resistance genes in these *H. parasuis* isolates from Chinese pig farms (Table 5). Other genes were not detected in this study.

## Mutations in QRDRs of *gyr*A, *gyr*B, *par*C, and *par*E

We also identified several QRDR mutations among the resistant *H. parasuis* strains. Mutations in *gyrA* (S83F/Y, D87Y/N/H/G) were detected in 116 (81%) of the strains. In addition, 79 strains had *parC* mutations (L379I/ Y557C/ V648I/E678D) and most of these were accompanied by *gyrA* mutations. Only nine strains had single *parC* mutations that were either L379I, Y557C, E678D, L379I or Y557C. The strains with *gyrA* mutations at either codon 83 or 87 showed higher MIC values compared with the 18 strains lacking

**Table 4 Prevalence of ARG types isolated from *H. parasuis*.**

| Gene | Number identified | Prevalence (%) |
| --- | --- | --- |
| *ermA* | 1 | 0.70 |
| *ermB* | 1 | 0.70 |
| *catl* | 10 | 6.99 |
| *flor* | 13 | 9.09 |
| *tetB* | 34 | 23.78 |
| *tetC* | 5 | 3.50 |
| *rmtD* | 1 | 0.70 |
| *rmtB* | 17 | 11.89 |
| *aadA1* | 4 | 2.80 |
| *aac(3′)-* II*c* | 6 | 4.20 |
| *sul1* | 9 | 6.29 |
| *sul2* | 2 | 1.40 |
| $bla_{TEM-1}$ | 9 | 6.29 |
| $bla_{ROB-1}$ | 7 | 4.90 |

**Table 5 Resistance gene patterns and the number of resistant strains.**

| Pattern | No. of isolates | Pattern | No. of isolates |
| --- | --- | --- | --- |
| *ermA* | 1 | *flor+aadA1* | 1 |
| *ermB* | 1 | *catl1+tetB* | 3 |
| *tetB* | 16 | *rmtB+ aac(3′)-IIc* | 1 |
| *catl1* | 1 | $rmtB+ bla_{TEM-1}$ | 3 |
| *tetC* | 1 | *rmtB+sul1* | 4 |
| *rmtB* | 6 | *sul1+ aac(3′)-IIc* | 2 |
| *flor* | 4 | $sul2+ bla_{ROB-1}$ | 1 |
| *sul1* | 2 | *sul2+tetB* | 1 |
| *aadA1* | 1 | $bla_{ROB-1}+aadA1$ | 1 |
| *aac(3′)-IIc* | 1 | $catl1+tetB+ bla_{TEM-1}$ | 1 |
| $bla_{TEM-1}$ | 4 | $catl1+tetB+ bla_{ROB-1}$ | 2 |
| $bla_{ROB-1}$ | 1 | *catl1+tetB+flor* | 1 |
| $catl1+bla_{ROB-1}$ | 1 | *catl1+tetB+aac(3′)-IIc* | 1 |
| *tetB+flor* | 2 | *tetB+flor+rmtB* | 1 |
| *tetB+aadA1* | 1 | *tetB+flor+ aac(3′)-IIc* | 1 |
| $tetB+ bla_{ROB-1}$ | 1 | *tetB+tetC+flor* | 1 |
| *tetB+tetC* | 1 | $rmtD+rmtB+ bla_{TEM-1}$ | 1 |
| *tetC+flor* | 1 | *tetB+tetC+flor+rmtB+sul1* | 1 |

**Table 6** QRDR mutations and antibiotic MIC values for 143 *H. parasuis* isolates.

| QRDR mutation | | Number of strains | MICs (µg/mL) | | | | | |
|---|---|---|---|---|---|---|---|---|
| *gyrA* − | *parC* − | 18 | Nalidixic acid 0.25–128 | Levofloxacin <0.25–2 | Ciprofloxacin <0.25–4 | Enrofloxacin <0.25–2 | Norfloxacin <0.25–4 | Lomefloxacin <0.25–1 |
| S83F/Y | − | 8 | 1–>512 | 0.25–16 | 0.25–16 | <0.25–8 | 0.25–256 | <0.25–4 |
| S83F/Y, D87Y/N/H/G | − | 38 | 4–>512 | 0.25–32 | 1–>512 | 0.25–32 | 0.25–>512 | 0.25–64 |
| S83F/Y | [a]L379I/Y557C/ V648I | 20 | 32–>512 | 0.25–64 | 0.25–32 | <0.25–32 | 0.25–16 | <0.25–128 |
| D87Y/H | [b]L379I/Y557C | 2 | 4, 16 | 0.25, 0.5 | 0.25, 0.5 | 2 | 1, 4 | 0.25, 0.5 |
| S83F/Y, D87Y/N/Y/G/H | [c]L379I/Y557C/ V648I/E678D | 48 | 1–>512 | 2–128 | 2–64 | 0.25–32 | 0.25–>512 | <0.25–64 |
| − | [d]L379I/Y557C/ L379I, Y557C, E678D/L379I, Y557C | 9 | 0.5–>512 | 0.25–8 | 0.25–16 | 0.5-16 | 0.25–>512 | 0.5–64 |

Notes.
  Mutation mode
[a]L379I; L379I+ Y557C+V648I; Y557C+ V648I; L379I+ Y557C; L379I+V648I.
[b]L379I +Y557C; L379I.
[c]L379I; Y557C; L379I+Y557C+V648I; L379I+Y557C+E678D; L379I+Y557C; Y557C+V648I.
[d]L379I; Y557C; L379I+ Y557C+E678D; L379I+Y557C.

mutations. The MIC values of the strains with single *parC* mutations were not significantly different from controls. No mutations were found in *gyrB* and *parE* (Table 6).

## PFGE
The 73 *H. parasuis* strains carrying resistance determinants were typed by PFGE and were genomically heterogenic. We identified 51 unique *CpoI* patterns but no evidence of clonality (Fig. 1).

## DISCUSSION
In the current study, we observed high-level resistance to nalidixic acid and enrofloxacin. Similar results have been reported such as 84.8% to nalidixic acid (*Xu et al., 2011*) and 60.1% and 45.5% to enrofloxacin (*Xu et al., 2011*; *Zhang et al., 2013*). These differed from results in the United Kingdom and Spain (0 and 20%) (*De la Fuente et al., 2007*). We described the fluoroquinolone antimicrobial resistance profiles for *H. parasuis* strains isolated between 2014–2017. When compared with 2002–2009 and 2008-2010, our data indicated that fluoroquinolone antimicrobial resistance in *H. parasuis* was very serious in China during the last 15 years.

There have been few complete and systematic molecular studies of antimicrobial resistance in *H. parasuis*. The genes $bla_{ROB-1}$, *tetB*, *tetL*, *qnrA1*, *qnrB6*, *aac* (6′)-*Ib-cr*, *lnu(C)* and *flor* were the only that were previously identified and that correlated with resistance (*Dayao et al., 2016*; *Guo et al., 2011*; *Kehrenberg et al., 2005*; *Lancashire et al., 2005*; *Li et al., 2015*; *San Millan et al., 2007*). Cephalosporinases, which are naturally present in some enterobacterial species, can be mobilized by transposons and migrate *via* plasmids
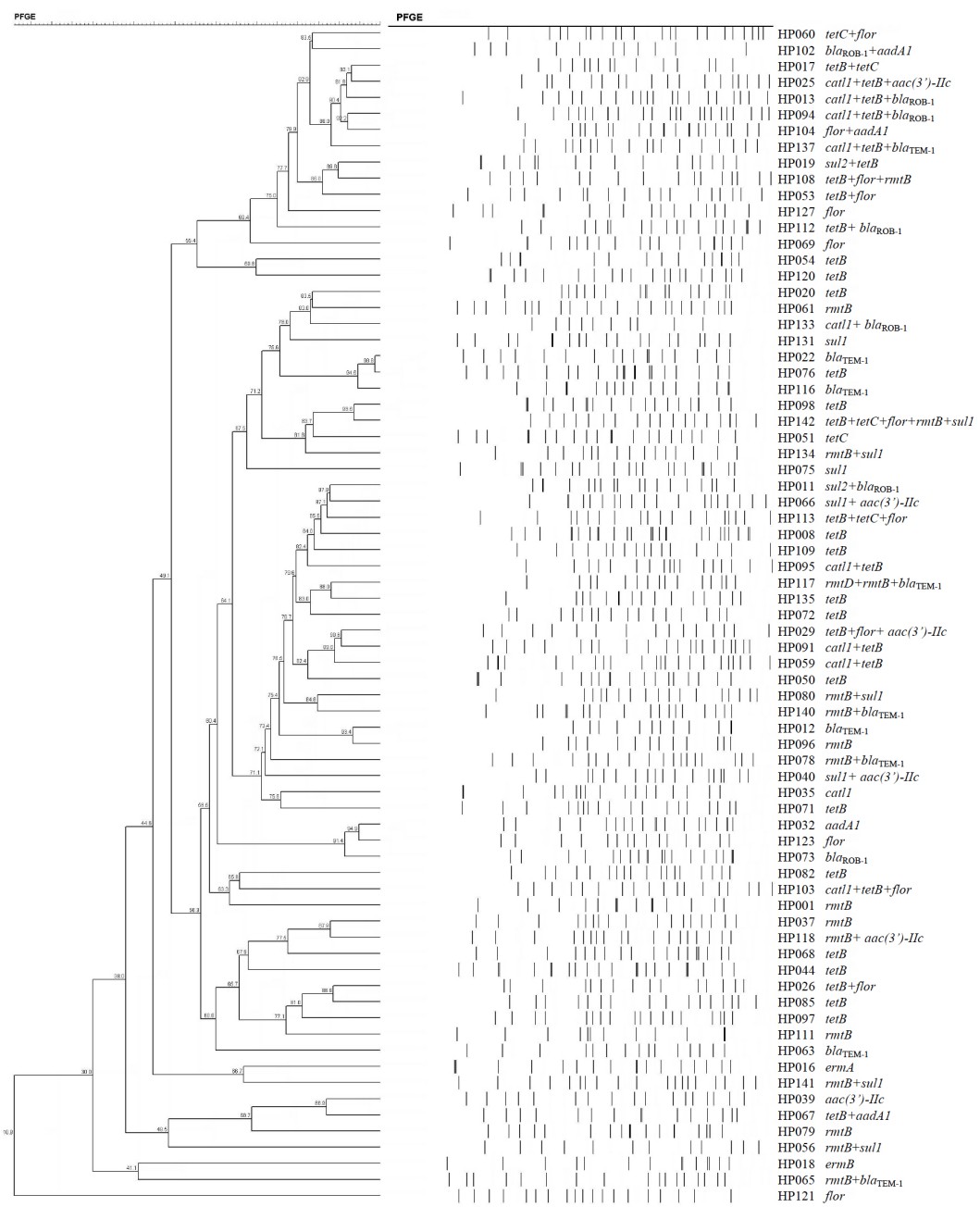

**Figure 1** Dendrogram of patterns generated by PFGE of 73 ARG-containing *H. parasuis* isolates

into other species. Moreover, the abuse of antimicrobial agents increases the number of carbapenem-resistant strains generating a public health concern (*Yang et al., 2017*). In the Enterobacteriaceae, the $bla_{TEM-1}$ β-lactamase is the predominant genotype (*Yang et al., 2017*). In our study, we identified both $bla_{TEM-1}$ and $bla_{ROB-1}$ β-lactamase genes which are widespread among *H. parasuis* and *Pasteurella spp* (*Guo et al., 2012*; *San Millan et al., 2007*). $bla_{TEM-1}$ and $bla_{ROB-1}$ are usually present in *H. influenzae* and have particular geographic

distributions in different countries (*Farrell et al., 2005*). These geographic differences may also be present in *H. parasuis*. The first reports of $bla_{TEM-1}$ and $bla_{ROB-1}$ were in China and Spain, respectively(*Guo et al., 2012*; *San Millan et al., 2007*). $bla_{ROB-1}$ was located on plasmid pB1000 and recently a novel 2,661 bp plasmid (pJMA-1) bearing $bla_{ROB-1}$ has been identified. This plasmid possessed a backbone found in small *Pasteurellaceae* plasmids and was 100% stable with a lower biological cost than pB1000 (*Moleres et al., 2015*).

We also identified genes encoding tetracycline efflux pumps (*tetB* and *tetD*) in this study. The first tetracycline resistant gene identified in *H. parasuis* was *tetB* and this gene is the most common tetracycline resistance gene in *Actinobacillus pleuropneumoniae* and *Pasturella multocida* (*Dayao et al., 2016*; *Matter et al., 2007*). The genes *tetH* and *tetM* are present in other members of the *Pasteurellaceae* (*Roberts, 2012*). Furthermore, the *tetB*-carrying plasmid pHS-Tet in *H. parasuis* was similar to a *tetL*-carrying plasmid in *Pasteurella* isolates (*Kehrenberg et al., 2005*; *Lancashire et al., 2005*). This is the first report of the *tetD* gene in *H. parasuis* isolates from China and needs further study. Tetracycline resistance genes are often associated with conjugative and mobile genetic elements enabling horizontal transfer (*Dayao et al., 2016*; *Roberts, 2012*). The presence of *tetD* suggests that tetracycline resistance in *H. parasuis* relies on efflux pumps.

In bacteria with animal origins, five florfenicol resistance genes (*floR, fexA, fexB, cfr* and *optrA*) have been reported (*Schwarz et al., 2004*; *Wang et al., 2015*). In Gram-negative bacteria, *floR* makes the greatest contribution to florfenicol resistance and this has been described for a number of bacterial species (*He et al., 2015*; *Meunier et al., 2010*; *Schwarz et al., 2004*; *Wang et al., 2015*). The emergence of florfenicol resistance in *H. parasuis* isolates was attributable to a novel small plasmid pHPSF1 bearing *floR*. This novel plasmid was similar to other *Pasteurellaceae* plasmids suggesting these species prefer to exchange genetic elements with each other.

High-level aminoglycoside resistance mediated by the production of the 16S rRNA methylases *armA, rmtA* to *H* and *npmA*, and resistance is increasing among Gram-negative pathogens (*Du et al., 2009*), being sometimes clonal spread of a single pulsotype (*Hopkins et al., 2010*). In our case, a clone bearing *rmtB* HP118 and HP037, was present in two different regions. However, until now, few studies have described the presence of the *armA* and *rmtB* genes in *H. parasuis* isolates, although they have been frequently reported on Enterobacteriaceae from food animals. The strains in our study also carried *rmtB, rmtD, aadA1* and *aac (3′) IIc* and these warrants further investigation.

The macrolide-resistance genes *erm A and erm B* showed a low frequency in our *H. parasuis* isolates. These genes are responsible for ribosomal binding site modifications that are the most important macrolide resistance mechanisms(*Takaya et al., 2010*).

The *sul1, sul2* and *sul3* genes are dihydropteroate synthases involved in sulfonamide resistance of Gram-negative bacteria and are usually associated with an integron system and a conjugative plasmid (*Vo et al., 2006*). In the current study, we identified both *sul1* and *sul2*, and these genes most likely accounted for the observed resistance to trimethoprim-sulfamethoxazole. These results are similar to others in Gram-negative bacteria (*Koljalg et al., 2009*; *Matter et al., 2007*).

This is the first report describing the presence of the *tetC, sul1, sul2, ermA, ermB, catl, rmtB, rmtD*, *aadA1* and *aac (3′)-IIc* genes in *H. parasuis,* to the best of our knowledge. Nevertheless, we did find several isolates with reduced antibiotic susceptibility that did not harbor any of the tested resistance genes. This suggests that *H. parasuis* possesses other resistance mechanisms such as mutations, decreases in permeability and increases in efflux pump activity or yet unknown antibiotic resistance mechanisms. In addition, the widespread dissemination of resistance genes and integrons could potentially fuel the rapid development of antimicrobial resistance due to their high transfer capabilities (*Hussein et al., 2009*). Therefore, more study is needed on this subject.

There have been numerous studies demonstrating *gyrA* and *parC* mutations engendering fluoroquinolone resistance in Gram-negative bacteria and Gram-positive bacteria from pigs such as *Salmonella spp., E. coli* or *Streptococcus suis* (*Cao et al., 2017*; *Escudero et al., 2007*). In *H. parasuis*, the *gyrA* mutations S83Y, S83F, D87Y, D87N and D87G are correlated with fluoroquinolone resistance. In addition, the *parC* mutations Y577C, V648I, E678D, S669F, A464V and A466S and *parE* mutations S283G, A227T and G241S were also found in these strains(*Guo et al., 2011*). In another study, mutations of *gyrA* D87N, *parC* S73R and *parE* T551A were involved in fluoroquinolone resistance, but other mutations such as in *gyrA* (452D^V/G, 627G^E), *gyrB* (211V^I, 254D^G), *parC* (73S^R/I, 227Q ∧H, 379L^I, 578C^Y) and *parE* (551T^A) occurred less frequently (*Zhang et al., 2013*). However, the *parE* mutation in *A. pleuropneumoniae* is possibly not involved in enrofloxacin resistance (*Wang et al., 2010*). In our study, most strains possessed *gyrA* mutations, and six strains possessed a *gyrA* mutation (D87H) not been previously reported. However, we do not know whether this mutation is directly related to fluoroquinolone resistance. We also identified four *parC* mutations. Unlike other studies, we found the *parC* 578 mutation in both resistant and sensitive strains, suggesting this mutation is not involved in resistance (*Zhang et al., 2013*). Overall, the QRDR analysis in our study suggested that the mutations at codon 83 or 87 of *gyrA* were responsible for fluoroquinolone resistance and that *gyrB* and *parE* were not.

Interestingly, our PFGE results indicated that almost 70% of our *H. parasuis* were genetically diverse, similar to a recent report (*Guo et al., 2012*). These results are in contrast to a previous study presenting evidence for the clonal spread of β-lactam resistance (*San Millan et al., 2007*). Our data suggests that resistance genes are spread *via* transferable elements such as plasmids and transposons in addition to clonal spread. Therefore, research on mechanisms for the spread of antimicrobial resistance in *H. parasuis* needs further investigation.

## CONCLUSIONS

In this study, we comprehensively and systematically investigated for the first time the distribution of the most common resistance genes in *H. parasuis* in China. These genes included *tetB, tetC, sul1, sul2, ermA, ermB,* $bla_{TEM-1}$, $bla_{ROB-1}$*, catl, flor, rmtB, rmtD, aadA1* and *aac (3′)-IIc*. The *gyrA* mutations S83F/Y and D87Y/N/H/G correlated with fluoroquinolone resistance in *H. parasuis*. These strains were also genetically diverse as

judged by PFGE. These data suggest that antimicrobial resistance in *H. parasuis* is primarily the result of transferable determinants and multiple target gene mutations. The exact roles for these detected resistance determinants in *H. parasuis* await further study.

## ACKNOWLEDGEMENTS

We thank members of our laboratories for fruitful discussions.

### Funding

This work was supported by the National Natural Science Foundation of China (No. 31372479), Science and Technology Planning Project of Guangdong Province, China (No. 2015B090901059), and the National Key Research Program of China (grant 2017YFD0501404). The funders had no role in study design, data collection and analysis, decision to publish, or preparation of the manuscript.

### Grant Disclosures

The following grant information was disclosed by the authors:
National Natural Science Foundation of China: 31372479.
Science and Technology Planning Project of Guangdong Province: 2015B090901059.
National Key Research Program of China: 2017YFD0501404.

### Competing Interests

Lili Guo is an employee of Qingdao Yebio Biological Engineering Co., Ltd.

### Author Contributions

- Yongda Zhao conceived and designed the experiments, prepared figures and/or tables, authored or reviewed drafts of the paper, approved the final draft.
- Lili Guo performed the experiments, analyzed the data, prepared figures and/or tables, authored or reviewed drafts of the paper, approved the final draft.
- Jie Li performed the experiments, analyzed the data, authored or reviewed drafts of the paper, approved the final draft.
- Xianhui Huang contributed reagents/materials/analysis tools, prepared figures and/or tables, authored or reviewed drafts of the paper, approved the final draft.
- Binghu Fang conceived and designed the experiments, contributed reagents/materials/-analysis tools, authored or reviewed drafts of the paper, approved the final draft.

### Animal Ethics

The following information was supplied relating to ethical approvals (i.e., approving body and any reference numbers):

The animal research committees of the South China Agriculture University granted approval for this research.

## Data Availability

The raw data are provided in a Supplemental Information 1.

## Supplemental Information

Supplemental information for this article can be found online at http://dx.doi.org/10.7717/peerj.4613#supplemental-information.

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
