# Peer review of "Characterization of antimicrobial resistance genes in Haemophilus parasuis isolated from pigs in China"

_PeerJ, doi:10.7717/peerj.4613_

## Round 0.1 · original submission · Major Revisions

Please make substantial revisions based on the comments from the reviewers. Please also edit your English by a native speaker.

Reviewer 1 ·

Basic reporting

This study screened multiple antimicrobial resistance genes by PCRs in 143 Haemophilus parasuis isolates in piglets from China. The molecular epidemiology of isolates was also analyzed.
This study is a purely epidemiological survey, and the information provided is limited. The manuscript was not prepared well. Data presentation (in the text and especially in tables) and discussion should be more concise and effective. The English language needs a thorough check, preferably by a native speaker.

Experimental design

no comment

Validity of the findings

no comment

Additional comments

Many sentences are difficult to understand and many parts need rephrasing for example but not limited to
Abstract
line 14: "Haemophilus parasuis is a common porcine respiratory disease". Haemophilus parasuis is a pathogen but not a disease.
line 23: "blaTEM-1" not "TEM-1". Resistance genes "rmtD, aadA1, aac(3’)-ⅡC" should be writtern in italic. Similar errors exist in Supplemental S2.
line 27: "pulsed-field gel electrophoresis". The letter "p" should be capitalized.
line 30: "high-level antibiotic resistance" would be better.
line 32: The author stated that "GyrA gene mutation also was the most important role in quinolone resistance". Actually, whether these gyrA mutations in this study cause quinolone resistance has not been validated.

Materials and Methods
line 71: A total of 143 or One hundred and forty-three.
line 92: change including to mediating

Reviewer 2 ·

Basic reporting

Several grammatical shortcomings have been identified throughout the manuscript. I suggest the manuscript should be revised by a native English speaker.

Experimental design

Some additional tests should be added, for example S1-PFGE and plasmid conjugation.

Validity of the findings

no comment

Additional comments

The manuscript describes antimicrobial resistance genes in H. parasuis isolated from pigs by PCR and pulsed-field gel electrophoresis. Some interesting resistance genes such as rmtB, blaROB, and qnr were found, which probably lead to the high antibiotic resistance of H. parasuis in piglets, but all of them lacked in-depth analysis. The reported data may be interesting for the readership. However, major revision need to be made before it will be accepted for publication.


The amendments are as follows,
Several grammatical shortcomings have been identified throughout the manuscript. I suggest the manuscript should be revised by a native English speaker.

In sections of the methods, “Bacterial isolates”: please add a brief description about how to obtain the samples and the detailed methods for isolation the strain.

In sections of the methods, if possible, the authors should study the location and horizontal transfer of some of the important antimicrobial resistance genes.

In the results section, antibiotic MIC values for 143 H. parasuis isolates should list the detailed data.

The following specific comments also need to be considered in the revised version of the current manuscript.
• The resistance-related protein and antimicrobial resistance genes need to be discriminated, such as ROB-1 to blaROB-1 and TEM-1 to blaTEM-1.

• Line 36: Abbreviations should be defined when they first appear, such as “Haemophilus parasuis is the etiological agent of Glässer’s disease that….” to “Haemophilus parasuis (H. parasuis) is the etiological agent of Glässer’s disease that….”.
• Line 71: Paragraphs should not begin with numbers. Please, rephrase the follow sentence: “143 H. parasuis strains were isolated from diseased swine suffering ….”.
• Line 152-153: The name of bacterium should be in italic type.
Tables and Figures
• Table 4 This table is too long, remove the Isolate column, re-design the table.
• Table 5 The expression forms of MICs should be unified. Please, “——, —” by “-”; The unit of MICs should use μg/mL.

·

Basic reporting

The manuscript is scientifically sound, professional and interesting. It reports new findings, that merit publication.
Figures and Tables should be clarified as stated in my report, e.g. Supplemental Material should be a Table in the manuscript.

Experimental design

No concerns at all. The study is well designed. Information of the antimicrobial susceptibility of all isolates should be included, if available.

Validity of the findings

Data are robust and relevant. The presentation of the results is somewhat confusing. With suggestions about Figure 1 and Supplemental Table, linking the data, will very much help interpretation of the results.

Additional comments

The manuscript by Zhao et al -Characterization of antimicrobial resistance genes in Haemophilus parasuis isolated from pigs in China- is of great scientific relevance and very interesting. Some clarification in the data would largely facilitate the Reading and interpretation of the results.

Has an antibiogram been performed for all the 143 isolates? Only 73 had one of the genes tested? The identification of the gens is interesting, but one should know if the genes are expressed and confer resistance or not.
No integrons have ever been identified in Pasteurellaceae. Please do not mention or discuss this, as in L212.

Do the mutation in the QRDR correspond to presence/absence of qnr genes?

Table S2 is much mor important and informative than Tables 1 and Table 2. Please include Table 3 in the main text. Otherwise the Pulsotype in Figure 1 is not interpretable.

Figure 1. Include a column on the right of the strain name with the genes present in each isolate. This will help the reader to identify clonality of the isolates.

Minor comments: L217 (…) in Gram negative and Gram-positive bacteria from pigs such as Salmonella spp., E. coli or Streptococcus suis (Cao et al 2017, Escudero JA et al Antimicrob Agents Chemother. 2007 Feb;51(2):777-82.)

L182 (…) and resistance is increasing among Gram-negative pathogens (Du et al 2009), being sometimes clonal spread of a single pulsotype (Hopkins KL et al. Emerg Infect Dis. 2010 Apr;16(4):712-5. ). In our case, a clone bearing rmtB HP118 and HP037, was present in two different regions.

---

## Round 0.2 · Minor Revisions

Please further improve your manuscript based on the reviewer's comments.

Reviewer 2 ·

Basic reporting

The authors have addressed most comments. But, it still needs attention in some places, for example, in Table 1, resistance genes should be written in italic. In addition, please add the information on how to identify H. parasuis. Other suggestions are presented as the following.

Experimental design

No concerns at all. if possible, the information on how to identify H. parasuis. should be added.

Validity of the findings

no comment.

Additional comments

(1) For Abstract, no information was on description of fluoroquinolone antimicrobial susceptibility testing in Methods and Results.
(2) Lines 70-71: “Verbal informed consent was obtained from the owners of the pigs before this study” should be deleted.
(3) Lines 115-116: “Information of the fluoroquinolone antimicrobial susceptibility of all isolates are listed in supplemental S1. The results of fluoroquinolone antimicrobial susceptibility testing of 143 H. parasuis isolates towards six fluoroquinolone antibiotics showed that 82.52% were resistant to nalidixic acid and 55.94% were resistant to enrofloxacin.”
It could be changed into “the results of the fluoroquinolone antimicrobial susceptibility of 143 H. parasuis isolates are listed in supplemental S1. It showed that 82.52% and 55.94% of all isolates were resistant to nalidixic acid and enrofloxacin, respectively.”
(4) Lines 196-197: “However, until now, few studies have described the presence of the armA and rmtB genes in food animals.”
It could be changed into “However, until now, few studies have described the presence of the armA and rmtB genes in H. parasuis isolates, although they have been frequently reported on Enterobacteriaceae from food animals.”
(5) Table 1, 2 and 3 are not very useful and can be supplementary files.
(6) Table 1,
1) “TEM” represents the name of enzyme. “blaTEM ” represents the name of resistance gene. “TEM” should be changed into blaTEM”, please checked other genes.
2) “fosA3” confers resistance to fosfomycin, not macrolids.

---

## Round 0.3 · accepted · Accept

Thank you for choosing PeerJ. We look forward to your future consideration of publishing in PeerJ.

#